# Prevalence and associated factors of microbial water quality from drinking water in Ethiopia: A systematic review and meta-analysis protocol

**Asmamaw Deguale Worku** [1,2]*, **Bezatu Mengistie Alemu** [1]

**1** Department of Water and Health, Ethiopian Institute of Water Resources, Addis Ababa University, Addis Ababa, Ethiopia, **2** Department of Public Health Emergency Management, Addis Ababa Health Bureau, Addis Ababa, Ethiopia

* asmamawdeguale16@gmail.com

**Editor:** D. Daniel, Gadjah Mada University Faculty of Medicine, Public Health, and Nursing: Universitas Gadjah Mada Fakultas Kedokteran Kesehatan Masyarakat dan Keperawatan, INDONESIA

## Abstract

### Purpose

To determine the pooled prevalence and associated contributing factors of microbial water quality, especially the most recent fecal contamination indicator, fecal coliforms, from drinking water in Ethiopia.

### Method

The review will be conducted per the Preferred Reporting Items for Systematic Reviews and Meta-Analyses (PRISMA) guidelines and PRISMA-P and registered in PROSPERO CRD42024537804. The studies will be identified from peer-reviewed literature, grey literature, and expert submissions. To identify peer-reviewed literature, "microbial water quality" will be combined with terms to restrict the search for drinking water and measure the prevalence and/or associated factors of microbial water quality. We further restricted the search in Ethiopia. The following databases will be used: PubMed/MEDLINE, Google Scholar, Worldwide Science, and Science Direct. Two independent reviewers will identify studies, extract data, assess the risk of bias, and assess methodological quality. The studies included will be determined in terms of quality based on the criteria listed in the Goanna Bridge Institute quality parameters. Statistical techniques like Higgins $I2$ will be used to investigate heterogeneity among the included studies. Sensitivity and subgroup analysis will be carried out to evaluate how reliable the results are. A funnel plot will be used to evaluate reporting publication bias, and Begg's and Egger's assessments will be used to check funnel plot balances.

### Discussion

This review and meta-analysis will thoroughly discover and integrate the data available on the prevalence and associated factors of fecal coliforms contamination in drinking water. The findings from this systematic review and meta-analysis will be compared and discussed with those from other studies.

**Data availability statement:** No datasets were generated or analyzed from this current study protocol.

**Funding:** The author(s) received no specific funding for this work.

**Competing interests:** The authors have declared that no competing interests exist.

**Abbreviations:** CFU, Colony Forming Unity; *E.Coli*, Escherichia Coli; GBI, Goanna Bridging Institute; JMP, Joint Monitoring Program; PRISMA, Preferred Reporting Items for Systematic Reviews and Meta-Analyses; ROB, Risk of Bias; TTCs, Thermo Tolerant Coliforms; UN, United Nation; UNICEF, United Nations Children's Fund; USA, United States of America; WHO, World Health Organization.

## Conclusion

Our systematic review and meta-analysis will help to develop specific recommendations for identified fecal coliforms contamination and associated factors in drinking water in Ethiopia. Moreover, this study will identify research gaps and guide future research and public health measures.

## Introduction

Drinking water contamination is a worldwide problem and causes a great threat to public health. Drinking water can be contaminated at the source, distribution system, and point of use, and such contaminated water can be a route for many microorganism transmissions [1,2]. Approximately 70% of the earth's crust is covered in water, but the majority of these sources are not fit for human consumption; just 2.5–3% of the water on land is drinkable [3,4]. Drinking water availability is not only essential for well-being but also for acceptable development, food security, and the improvement of living conditions. Safe and clean drinking water were recognized as fundamental human rights by the UN General Assembly on July 28, 2010, by Resolution A/RES/64/292 [5–8]. These rights are necessary for the full enjoyment of life and all other human rights [9,10].

According to WHO/UNICEF's Joint Monitoring Programme for Water Supply and Sanitation (JMP), an enhanced drinking-water source type is one that is "protected from outside pollution, "in particular against contamination with fecal matter, by nature of its architecture or through forceful intervention" [8,11]. Rainwater, protected springs or wells, standpipes, boreholes, and piped water into homes, yards, or plots are examples of improved source types [12]. Unprotected wells, unprotected springs, surface waterways, and tanker trucks are examples of unimproved source types that do not shield water from external contamination [12,13]. Although the classification adheres to accepted standards for hygienic protection, the JMP issued a warning when declaring in 2010 that the target had been reached because water quality measurements are not included in the Millennium Development Goals indicator, and many improved and unimproved water sources were potentially contaminated by fecal matters [13–15].

Globally, 1.8 billion people utilize a drinking water source that is polluted by feces, from this, 1.1 billion drink water is of at least 'moderate' risk (>10 *Escherichia coli* or thermo tolerant coliform per 100 ml) [16,17]. Drinking water is found to be more polluted in rural (41%) than urban (12%) areas, and pollution is predominant in Africa (53%), and South-East Asia (35%) [18,19]. Fecal coliform bacteria contamination of drinking water in rural community exerts a great public health problem in low and middle income countries [20]. Several studies also showed that drinking water were contaminated by fecal coliforms like in Nepal 81% at house hold stored container and 68% at point of collection [19], 94% of drinking water in Myanmar was contaminated with TTCs, ranging from 2.2 CFU/100 mL to more than 1000 CFU/100 mL [21], the highest level of total coliform (TC) was recorded 37.26 CFU100 ml and the lowest is 22.13 CFU/100 ml in India [22], 78.42% of drinking water sources in Pakistan were contaminated by fecal coliforms bacteria [23,24],other study in Pakistan also 80% drinking water sources were contaminated with sewerage (fecal) matter [25], in Nepal also 88.5% were positive for total coliform and 56.5% were positive for fecal coliform [26], in North Carolina 29.2% in private well water were positive for total coliforms bacteria and 6.43% for *E. coli* [27–29].

WHO 2017, guidelines about drinking water and water quality, as well as the Ethiopian Standard Agency guidelines, find that there is no place to detect coliform bacteria in 100-ml water samples [17,30,31]. In Ethiopia, 60% to 80% of the total population suffers from

waterborne and water-related diseases [16]. TTCs are commonly used as indicators of recent fecal contamination, which poses a risk to human health. *Escherichia coli, Klebsiella, Enterobacter,* and *Citrobacter* [32] collectively termed as "fecal coliforms or thermo tolerant coliforms", making, it possible to identify and assess potential health risks, but for the purpose of this study we used fecal coliforms or TTCs rather than each specific species [12].

Despite many hopeful activities and efforts made to ensure safe water supply to the people of the world in recent years, fecal contamination of drinking water remains a great health concern in developing countries [33] like in Ethiopia [2,34], Kenya [35], Indonesia [36,37], Peru [38], Pakistan [25,39], India [40], in China [41] and in other developing countries [42–44]. A study conducted in Vietnam to assess contributing factors of fecal coliform contamination of drinking water revealed that water from dug wells, collecting water stored in household tanks by a hand-held ladle, household size ≥ 5 members, and scooping water by a cup, a visually dirty container were risk factors for fecal contamination of household drinking water [45], in Ghana seasonal variation also the factors associated with fecal coliforms contamination of drinking water [46], fecal coliforms contamination of drinking water in China has been linked to seasonal variation and the physicochemical characteristics of the water [47]. Fecal contamination of drinking water in Rwanda was linked also to lower elevation, increased open waste disposal in a particular area, sources of water other than piped to households or rainwater/bottled water, and the frequency of intense rain events [48]. The presence of latrine, family size, income status, water shortage experience, and educational status were the predictors of fecal contamination of drinking water [49–51]. The prevalence of fecal coliforms and their associated factors in drinking water in Ethiopia is different in different parts of the country, in urban and rural areas. There is no precise pooled overall prevalence of fecal coliforms and its contributing factors in Ethiopia. Moreover, the present overall pooled prevalence of fecal coliforms in drinking water is not well known in this setup. Therefore, this systematic review and meta-analysis study aims to produce the pooled prevalence and contributing factors of fecal coliforms contamination in drinking water with available studies in Ethiopia.

### Review questions

1. What is the pooled overall prevalence of fecal coliforms contamination of drinking water in Ethiopia?

2. What are the contributing associated factors of fecal coliforms contamination of drinking water in Ethiopia?

## Method

The review will be conducted by the Preferred Reporting Items for Systematic Reviews and Meta-Analyses (PRISMA) guidelines as well as the PRISMA-P checklist [52,53] and registered in PROSPERO CRD42024537804.

### Search strategy

The studies will be identified from peer-reviewed literature, grey literature, and expert submission on drinking water. To identify peer-reviewed literature, "microbial water quality" will be combined with terms to restrict the search to drinking water and measure the prevalence and/or associated factors of microbial water quality. We further restricted the search in Ethiopia. The following databases will be used: PubMed/MEDLINE, Google Scholar, Worldwide Science.org, and Science Direct.

### Search terms

To increase the probability of all-inclusiveness of the findings, search terms were developed and combined using Boolean operators as follows:

The search terms used were "prevalence," "associated factors," "microbial water quality," "bacterial water quality," and "drinking water in Ethiopia. Prevalence and associated factors of microbial water quality or associated factors of bacterial water quality from drinking water in Ethiopia. We would undertake advanced searching by combining the search terms using Boolean operators.

### Search detail

(((("epidemiology"[MeSH Subheading] OR "epidemiology"[All Fields] OR "prevalence"[All Fields] OR "prevalence"[MeSH Terms] OR "prevalance"[All Fields] OR "prevalences"[All Fields] OR "prevalence s"[All Fields] OR "prevalent"[All Fields] OR "prevalently"[All Fields] OR "prevalents"[All Fields]) AND ("associate"[All Fields] OR "associated"[All Fields] OR "associates"[All Fields] OR "associating"[All Fields] OR "association"[MeSH Terms] OR "association"[All Fields] OR "associations"[All Fields]) AND ("factor"[All Fields] OR "factor s"[All Fields] OR "factors"[All Fields]) AND ("microbial"[All Fields] OR "microbially"[All Fields] OR "microbials"[All Fields]) AND ("water quality"[MeSH Terms] OR ("water"[All Fields] AND "quality"[All Fields]) OR "water quality"[All Fields])) OR (("associate"[All Fields] OR "associated"[All Fields] OR "associates"[All Fields] OR "associating"[All Fields] OR "association"[MeSH Terms] OR "association"[All Fields] OR "associations"[All Fields]) AND ("factor"[All Fields] OR "factor s"[All Fields] OR "factors"[All Fields]) AND ("bacterial"[All Fields] OR "bacterially"[All Fields] OR "bacterials"[All Fields]) AND ("water quality"[MeSH Terms] OR ("water"[All Fields] AND "quality"[All Fields]) OR "water quality"[All Fields]) AND ("drinking water"[MeSH Terms] OR ("drinking"[All Fields] AND "water"[All Fields]) OR "drinking water"[All Fields]) AND ("Ethiopia"[MeSH Terms] OR "Ethiopia"[All Fields] OR "Ethiopia s"[All Fields]))) AND ("2000/01/01"[PubDate]: "2024/03/11"[PubDate]).

All searches will be restricted to the years "between" 2000 and 2024 and only in the English language; however, each online database varies in terms of the earliest articles available online.

Grey literature Search terms will be modified in the databases where required. Basic terms will be used for the grey literature. Bibliographies of included studies and relevant reviews will be searched. To ensure the quality of the study, the gray literature we will review should be done by at least two authors. Based on the resources available, the titles and abstracts from selected journals will be reviewed. These papers will be selected based on the number of eligible articles identified in each paper in prior searches.

### Eligibility criteria for study selection

Studies will be included in the review based on the following criteria:

- The article contained extractable data on fecal coliforms bacteria or thermo-tolerant coliforms from any drinking water sources.

- The article was published in the years 2000 and 2024.

- Only observational study design articles such as cross-sectional, case-control, or cohort.

We will exclude the microbial water quality study that was not done in English and was outside of Ethiopia. We will include all studies that were conducted at all drinking water sources, either protected or unprotected wells, springs, taps, and any water sources that were used for drinking purposes. We will include articles on fecal coliforms studies only, instead of articles on specific species of fecal coliforms. WHO guidelines/Standards and the Ethiopian

standard agency for fecal coliform contamination of drinking water must not be detectable in any 100 ml sample [54–56].

## Study selection

The approach will differ for peer-reviewed and grey literature. Bibliographic software (End-Note X9) will be used to export all references from online databases of peer-reviewed research to a spreadsheet once duplicates are identified and removed. Study selection will be conducted in two stages: screening of titles and abstracts, followed by screening of full texts, as well as applying inclusion and exclusion criteria. The records will be screened for inclusion by two independent reviewers, ADW and BMA. Every reviewer will conduct their own independent assessment. If there are any discrepancies, the two reviewers will discuss and resolve them.

Grey literature will be screened by one reviewer. Search terms used in each database will be recorded, and the number of studies returned will be recognized along with the number selected based on the title and/or abstract. Selected studies were included in the spread sheet with the peer-reviewed literature, with an additional column to indicate their source of publication. The selection of study included in the review will be based on the following revised PRISMA diagram Fig 1 [57].

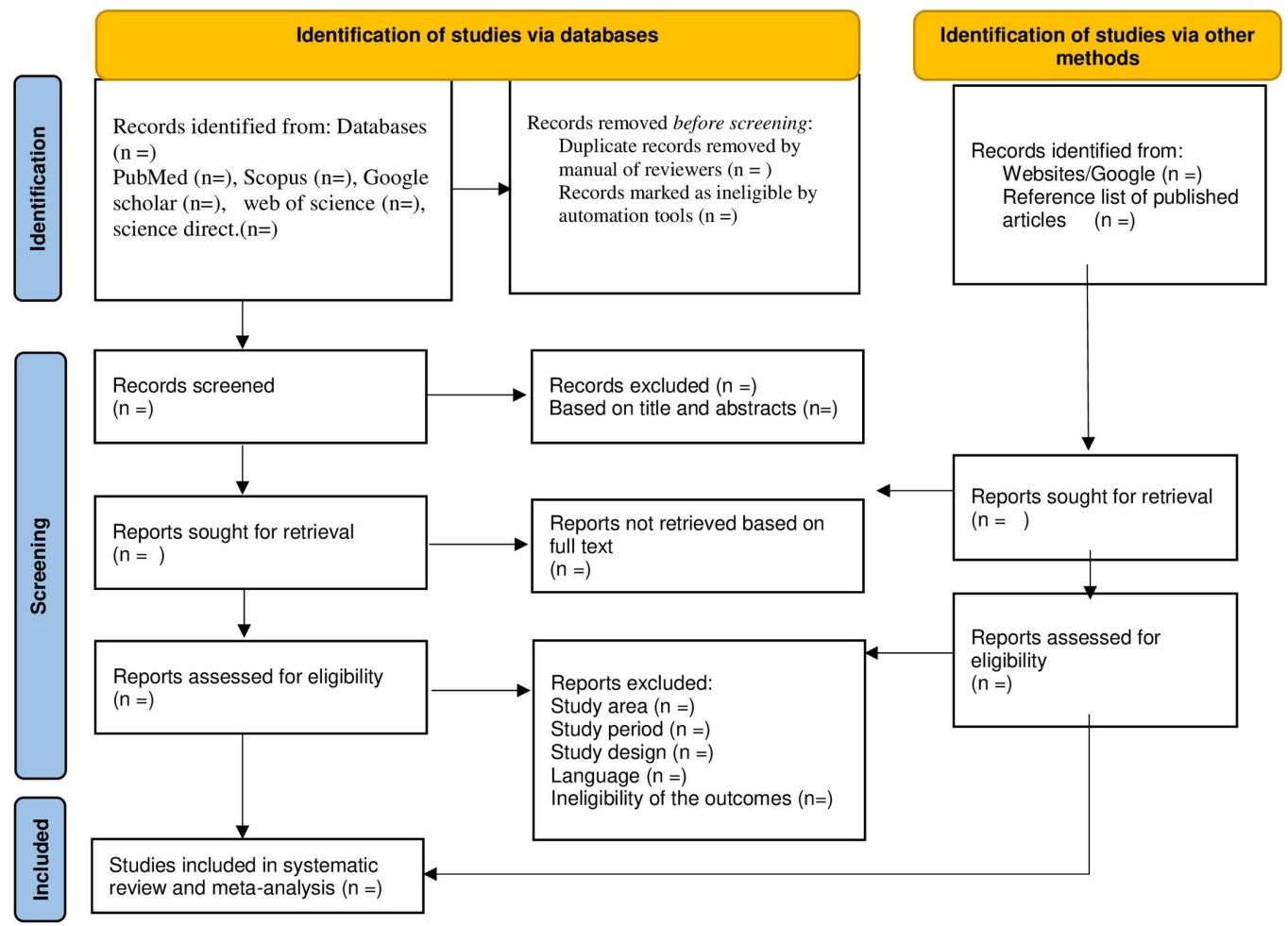

**Fig 1. Study selection procedure of flow diagram.**

## Data extraction and collection

A data extraction table will be developed in Excel to facilitate the extraction process. Descriptive data from eligible studies (author, year of publication, region, etc.) will be extracted, and we will also extract additional study characteristics thought to influence microbial water quality. Then characteristics of the study will be extracted, such as country, region, setting (urban/rural), study design, study duration, sample size), location of sample collection), odd ratio, confidence interval, p-value, parameters tested, quality score and percentage of sample in compliance with WHO guidelines or local context. We will classify studies based on study design, as this is thought to affect the extent to which they are affected by bias. Our classification includes cross-sectional, case-control, and cohort studies. Additional data will be extracted specific to the quality parameters that the study examined (prevalence and associated factors of fecal coliforms contamination in drinking water). Once the data extraction is completed, "10% of the included studies will be randomly selected from each reviewer to review separately" to assess the correctness of the data extracted, because complete texts would undergo the same review procedure [58]. The results from both reviewers (ADW and BMA) will be compared to determine the variability in data extraction and any disagreements will be resolved by agreement between the reviewers. The data from each included study is extracted in Table 1 [59] and the stage of the review during submission will be shown in Table 2 [59].

Data extraction table for systematic review and meta-analysis study.

## Regarding the missing data

The initial and/or corresponding authors will be contacted by phone, email, or other method to request any missing data and/or further information if any are not included in the included research. We will reach out the corresponding authors for missing element of the study up to three attempts to maximize the response rate.

## Heterogeneity

To know the source of heterogeneity, we use simple approaches to explore variations among all studies, and subgroups analysis and multiple meta-regression will be done based on data types that are used categorical data including the following characteristics, region (Amhara, Oromia, Tigrai…), study setting (urban vs rural), study design (cross-sectional, case-control and cohort),

**Table 1. Data extraction table for Characteristics of the studies included in the systemic review and meta-analysis.**

| Author | Publication Year | Publication Source | Region | Study design | Study setting | Study duration | Sample size | No of samples tested +(FC) | GBI quality score | Prevalence (%),(CI) | Associated factors OR (CI), or 2X2 classification of predictors. |
|---|---|---|---|---|---|---|---|---|---|---|---|
| | | | | | | | | | | | |

**Table 2. Stage of review at the time of submission.**

| Stage | Started | Completed |
|---|---|---|
| Preliminary searches | Yes | Yes |
| Piloting of the study selection process | Yes | Yes |
| Formal screening of search results against eligibility criteria | Yes | Yes |
| Data extraction | No | No |
| Risk of bias (quality) assessment | No | No |
| Data analysis | No | No |

source of publication (published and unpublished), and publication year (before 2015 and equal or more than 2015 until 2024) etc.), in alignment with our research objectives. Heterogeneity will be evaluated statistically by chi-square test (Q-test) statistics and inverse variance index ($I^2$). $I^2$ values will be classified as follows: low heterogeneity (0–25%), moderate heterogeneity (25–50%), and high heterogeneity (> 50%) [60,61]. We anticipate diverse results among the studies and will endeavor to understand the potential reasons for any specific outcomes. Forest plots will be produced to present the pooled estimates of the prevalence of fecal coliforms contamination in drinking water. Sub-groups have been identified, and random effect models will be applied before analyzing the data. We will compare results from different studies in different subgroups to identify disparities in microbial water quality across different groups.

## Assessment of publication bias

Studies with negative findings and, depending on the journal, interest could be published less frequently, thereby creating a publication bias. The extent of publication bias will be measured using a funnel plot and Egger's test for small study effects [62]. We will also assess sensitivity to observe the impact of individual studies on the overall results. We will also use Begg's test to quantify publication bias.

## Plans for updating the review

If necessary, revised or expanded versions of this systematic review and meta-analysis protocol will be made available.

## Ethical approval and consent to participate

This is not applicable because no primary data will be collected.

## Discussion

There is no previous study on the overall pooled prevalence and associated factors of fecal coliforms contamination of drinking water in Ethiopia. This review and meta-analysis will thoroughly discover and integrate the data available on the prevalence and associated factors of fecal coliforms contamination in drinking water [59,63]. In this review, evidence about the possible contributing factors for fecal contamination and its proportion in drinking water will be gathered, analyzed, and summarized. The review will be presented according to the PRISMA guidelines [64], and we will submit it to a suitable journal for publication. The findings from this systematic review and meta-analysis will be compared and discussed with those from other studies conducted to determine fecal coliforms contamination of drinking water and its contributing factors in different countries outside of Ethiopia like as Thailand [65,66], Cambodia [66], Nigeria [67], Ecuador [68], and in China [69].

The strength of this review is being reported according to the PRISMA-P statement [53,70], and the consequent systematic review and meta-analysis will be done according to PRISMA 2020 guidance [71]. Second, the review includes both peer-reviewed and non-peer-reviewed articles, which is the strength of the review. Thirdly, the review will utilize a comprehensive search strategy to increase the inclusiveness of the study. Potential limitation of this review is that it will include only English-language articles. The other potential limitation of this review is involvement of only two reviewers. If any discrepancy or disagreement occurs between the two reviewers and try to solve only between them, may result in poor quality of review process.

## Conclusion

Fecal contamination of drinking water is now a major public health problem in low- and middle-income countries, including Ethiopia, and causes many problems for human beings. Nevertheless, there is a dearth of evidence on the prevalence of fecal coliforms and its contributing factors to fecal contamination of drinking water. This systematic review and meta-analysis will deliver up-to-date data on the prevalence and associated factors of fecal coliforms contamination in drinking water. Our systematic review and meta-analysis will help to develop specific recommendations for identified fecal coliforms contamination and associated factors in drinking water in Ethiopia. Moreover, the results from this study will guide future research and public health measures with an understanding of the importance of microbial water quality and also point out directions for applying interventions to water safety.

## Supporting information

**S1 File. PRISMA-P[1] (Preferred Reporting Items for Systematic review and Meta-Analysis Protocols) 2015 checklist: recommended items to address in a systematic review protocol** [54]. (DOC)

## Author contributions

**Conceptualization:** Asmamaw Deguale Worku, Bezatu Mengistie Alemu.

**Data curation:** Asmamaw Deguale Worku.

**Formal analysis:** Asmamaw Deguale Worku.

**Investigation:** Asmamaw Deguale Worku.

**Methodology:** Asmamaw Deguale Worku.

**Project administration:** Asmamaw Deguale Worku.

**Resources:** Asmamaw Deguale Worku.

**Software:** Asmamaw Deguale Worku.

**Supervision:** Asmamaw Deguale Worku, Bezatu Mengistie Alemu.

**Validation:** Asmamaw Deguale Worku.

**Visualization:** Asmamaw Deguale Worku.

**Writing – original draft:** Asmamaw Deguale Worku.

**Writing – review & editing:** Asmamaw Deguale Worku.

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
