## [Decision Letter · Decision Letter 0]

19 Jun 2024

PONE-D-24-12272Prevalence and associated factors of microbial water quality from drinking water in Ethiopia: A Systematic Review and Meta-Analysis ProtocolPLOS ONE

Dear Dr. worku,

Thank you for submitting your manuscript to PLOS ONE. After careful consideration, we feel that it has merit but does not fully meet PLOS ONE’s publication criteria as it currently stands. Therefore, we invite you to submit a revised version of the manuscript that addresses the points raised during the review process.

We look forward to receiving your revised manuscript.

Kind regards,

D. Daniel, Ph.D.

Academic Editor

PLOS ONE

Journal Requirements:

-DOI https://doi.org/10.2147/JPR.S380058

In your revision ensure you cite all your sources (including your own works), and quote or rephrase any duplicated text outside the methods section. Further consideration is dependent on these concerns being addressed.

**Additional Editor Comments** : 

Please check comments from Reviewers and update your draft accordingly. Even though you state that this study protocol is for Ethiopia, but I think that the protocol should be sufficient to be implemented in different context/countries. So, you may cite other drinking water quality study in other regions outside Ethiopia (e.g., https://doi.org/10.3390/ijerph17072172 and https://doi.org/10.2166/washdev.2023.215) to make this protocol and your introduction broader. in short, expand a bit the introduction and discussion of your study protocol to broader context outside Ethiopia.

Please review and evaluate the requested works to determine whether they are relevant and should be cited. It is not a requirement to cite these works.

Reviewers' comments:

Reviewer's Responses to Questions

**Comments to the Author**

1. Does the manuscript provide a valid rationale for the proposed study, with clearly identified and justified research questions?

Reviewer #1: Partly

Reviewer #2: Yes

2. Is the protocol technically sound and planned in a manner that will lead to a meaningful outcome and allow testing the stated hypotheses?

Reviewer #1: Yes

Reviewer #2: Partly

3. Is the methodology feasible and described in sufficient detail to allow the work to be replicable?

Reviewer #1: Yes

Reviewer #2: Yes

4. Have the authors described where all data underlying the findings will be made available when the study is complete?

Reviewer #1: Yes

Reviewer #2: Yes

5. Is the manuscript presented in an intelligible fashion and written in standard English?

Reviewer #1: Yes

Reviewer #2: Yes

6. Review Comments to the Author

You may also provide optional suggestions and comments to authors that they might find helpful in planning their study.

**Reviewer #1:**  Prevalence and associated factors of microbial water quality from drinking water in

Ethiopia: A Systematic Review and Meta-Analysis Protocol

This study protocol examined the overall prevalence and predictors of fecal contamination in drinking water in Ethiopia. However, certain aspects of the materials and methods, including the study settings, sample sizes, and standard methods used for bacterial isolation, need to be clarified before publication.

Major criticism

- Was this study registered in PROSPERO?

- The term "household level of water" should be clarified.

- What automation tools were used in this study?

- The number and expertise of reviewers involved in article selection should be specified.

- How many reviewers are used for article selection? This point should be clarified.

- The rationale for selecting only 10% of the articles to assess data accuracy needs clarification.

- In Table 2, why will the language be included, since this study focuses only on the English language?

- The differences between study design, study setting, and study type need to be clearly defined.

- The factors/stratum used for the subgroup analysis should be identified.

- The term “microbial water quality” is broad; this study only reports the prevalence of fecal coliforms.

- The sources of drinking water should be specified, noting that water treatment affects bacterial contamination reduction.

- Standards/guidelines for fecal coliform contamination in drinking water should be included.

Minor criticism

- Change "Drinking water Availability" to "Drinking water availability".

- Ensure consistency in font usage.

- Italicize the names of bacterial species: Escherichia, Klebsiella, Enterobacter, and Citrobacter.

- Correct "Odd ratio" to "odds ratio".

- Change "Study time" to "duration of study".

- Specify the full name of the GBI quality parameter.

- Use "fecal coliforms" in plural form.

**Reviewer #2:**  This study protocol, entitled systematic review and meta-analysis on the prevalence and contributing factors of fecal coliform contamination in drinking water in Ethiopia addresses a critical public health issue. The methodology that was outlined is comprehensive and follows established guidelines, demonstrating a thorough approach to this important topic. However, I have identified several areas where the manuscript could be strengthened to enhance its clarity, depth, and overall impact. Please find below my detailed comments and suggestions for improvement.

The search terms should contain the species of fecal coliform bacteria, including Escherichia, Klebsiella, Enterobacter, and

Citrobacter. This can integrate all possible publications conducted in Ethiopia relevant to the exact microbial quality of drinking water in this country. In addition, in PRISMA flow chart, I recommend that publications that were not published in English be excluded at the first step of the screening process. Lastly, the criteria that will be used for subgroup analysis does not indicated in the protocol.

7. PLOS authors have the option to publish the peer review history of their article (what does this mean? ). If published, this will include your full peer review and any attached files.

**Do you want your identity to be public for this peer review?** For information about this choice, including consent withdrawal, please see our Privacy Policy .

Reviewer #1: No

Reviewer #2: No

---

## [Author Response · Author response to Decision Letter 1]

4 Jul 2024

Dear editor and reviewers,

Response: The authors are thankful for the detailed and specific comments on the protocol. As it is a protocol, we strongly believe that this will shape our direction for the systematic review to be conducted. Most comments focused on the introduction, methodology and discussion section as well as PLOS ONE's style of writing or requirements for publication. Therefore, detailed and extensive revision is made regarding introduction, methodology and discussion section as well as PLOS ONE's style requirements formatting, inclusion criteria, PRISMA flow diagram, heterogeneity, removing overlapping texts or duplicates, expand the introduction and discussion, supporting information, data availability and others. With all this revision, we believe that the protocol will enable us conduct a standard systematic review of scientific interest. Specific responses and revisions made are provided below and highlighted in the main document. Hopefully, the editor will consider the revised protocol for further review process.

Prevalence and associated factors of microbial water quality from drinking water in Ethiopia: A Systematic Review and Meta-Analysis Protocol

To: editor,

Above all, we thank you for your constructive suggestions and comments on this manuscript that would improve the substance and content of the protocol. We considered each comment and clarification question of editor on the manuscript thoroughly. Our point-by-point responses for each comment and question are described in detail on the following pages. All newly modified changes were highlighted within the main manuscript. Please see the red font colour in the clean revised manuscript.

Editor Comments and Suggestions

Please ensure that your manuscript meets PLOS ONE's style requirements, including those for file naming. The PLOS ONE style templates can be found at here.

Response: Thanks a lot for your eagle eyes! The editor is correct. We are very grateful for this implicit comment and suggestion. Thank you so much for such valuable intellectual input. Now, based on your suggestions and comments, we have thoroughly revised and edited all parts of our manuscript depending on PLOS ONE's style requirements. The newly modified change was highlighted with a red font color in the cleanly revised manuscript.

2. We noticed you have some minor occurrence of overlapping text with the following previous publication(s), which needs to be addressed. In your revision ensure you cite all your sources (including your own works), and quote or rephrase any duplicated text outside the methods section. Further consideration is dependent on these concerns being addressed.

Response: Thanks a lot for your eagle eyes! The editor is correct. We are very grateful for this implicit comment and suggestion. Thank you so much for such valuable intellectual input. Now, based on your suggestions and comments, we have thoroughly, cited, quoted, revised, paraphrased and edited all parts of our manuscript. The newly modified change was highlighted with a red font color in the cleanly revised manuscript.

Response: Thank you very much. The editor is perfect. We are grateful for your robust and rigorous comments and suggestions. Hence, the authors utterly share the concerns and comments of editor. Thus, now with great appreciation, the authors considered your valid concern and explicitly incorporation of data availability statement of the submission form during submission. The newly modified change was highlighted with a red font color in the cleanly revised manuscript

4. Please include captions for your Supporting Information files at the end of your manuscript, and update any in-text citations to match accordingly. Please see our Supporting Information guidelines for more information

Response: Thank you very much. The editor is perfect. We are grateful for your robust and rigorous comments and suggestions. Hence, the authors absolutely share the concerns and comments of editor. Thus, now with great appreciation, the authors considered your valid concern and explicitly incorporation of Supporting Information files at the end of manuscript with perfect captions and in-text citations. The newly modified change was highlighted with a red font color in the cleanly revised manuscript.

5. Additional Editor Comments:

Please check comments from Reviewers and update your draft accordingly. Even though you state that this study protocol is for Ethiopia, but I think that the protocol should be sufficient to be implemented in different context/countries. So, you may cite other drinking water quality study in other regions outside Ethiopia to make this protocol and your introduction broader. In short, expand a bit the introduction and discussion of your study protocol to broader context outside Ethiopia.

Response: Thanks a lot for your eagle eyes! The editor is correct. We are very grateful for this implicit comment and suggestion. Thank you so much for such valuable intellectual input. Now, based on your suggestions and comments, we have thoroughly expanded and broadened introduction and discussion section of the manuscripts by cited and included different drinking water quality study that conducted outside of Ethiopia, as well as revised and edited all parts of our manuscript. The newly modified change was highlighted with a red font color in the cleanly revised manuscript.

To Reviewers’

Above all, we thank you for your constructive comments and helpful suggestions that helped us to improve and enrich our protocol. Here in the below, we have pointed out how authors incorporated your valuable comments, suggestions and concerns one by one. Thank you so much.

Reviewer #1 General comment

1. Prevalence and associated factors of microbial water quality from drinking water in Ethiopia: A Systematic Review and Meta-Analysis Protocol

This study protocol examined the overall prevalence and predictors of fecal contamination in drinking water in Ethiopia. However, certain aspects of the materials and methods, including the study settings, sample sizes, and standard methods used for bacterial isolation, need to be clarified before publication.

Response: thank you very much; we believe that the study focused on fecal coliform contamination of drinking water in Ethiopia. Fecal coliforms are purely bacteria.

2. Was this study registered in PROSPERO?

Response: Sure, it is registered in PROSPERO CRD42024537804.

3. The term "household level of water" should be clarified.

Response: Definitely! The reviewer is correct. We are grateful for this implicit comment and suggestion. The authors critically reviewed this vital comment and corrected the necessary modification as per your robust and critical comments. The newly modified change was highlighted with red font color in the clean revised main manuscript

4. What automation tools were used in this study?

Response: Well thought comments and suggestions! The reviewer is perfect. The authors critically reviewed this vital comment and corrected the necessary modification as per your valuable and critical comments. The automation tool was reference manager software (EndNote X9).The newly modified change was highlighted with red font color in the clean revised main manuscript/PRISMA flow diagram.

5. The number and expertise of reviewers involved in article selection should be specified. How many reviewers are used for article selection? This point should be clarified.

Response: Thanks a lot for your eagle eyes! The reviewer is correct. We are very grateful for this implicit comment and suggestion. Thank you so much for such a valuable intellectual input. The newly modified change was highlighted with red font color in the clean revised main manuscript that incorporated two independent reviewers (ADW and BMA) were involved for article selection. Any discrepancy was resolved by discussion between two reviewers.

6. The rationale for selecting only 10% of the articles to assess data accuracy needs clarification.

Response: Thank you very much for your question. We are grateful for this implicit comment. To assess the correctness of the extracted data, take 10 % of the included study from each reviewer separately, due to the fact that complete texts would undergo the same review procedure. This is scientifically sound and used by different scholars.

7. In Table 1, why will the language be included, since this study focuses only on the English language?

Response: Thank you so much to pointing out this. The reviewer is perfect. We are very grateful for this implicit comment and suggestion. Thank you so much for such a valuable intellectual input. Authors critically considered this valid input. Now, the newly modified change was highlighted with red font color in the clean revised manuscript.

Data extraction table for systematic review and meta-analysis study

Author Publication Year Publication Source Region

Study design Study setting Study duration Sample size No of samples tested +( FC) GBI quality score

Prevalence (%),( CI) Associated factors

OR (CI), or 2X2 classification of predictors.

Table1: data extraction table for Characteristics of the studies included in the systemic review and meta-analysis

8. The differences between study design, study setting, and study type need to be clearly defined.

Response: Definitely! Many thanks for it. We are grateful for this crucial remark. Reviewer, you are flawless. This significant observation of highlights was thoroughly addressed in the updated paper. The newly modified change was highlighted with red font color in the clean revised manuscript.

9. The factors/stratum used for the subgroup analysis should be identified.

Response: Thank you so much for such valuable intellectual input. The reviewer is perfect. We are very grateful for this implicit comment and suggestion. Thank you so much for such valuable intellectual input. The factors that used for subgroup analysis was to detect the source of heterogeneity in the outcome variables. We used subgroup analysis and Meta regression if possible depending on the data types those is categorical type in this review protocol. The authors critically considered this valid input. The newly modified change was highlighted with a red font color in the cleanly revised manuscript.

10. The term “microbial water quality” is broad; this study only reports the prevalence of fecal coliforms.

Response: Many thanks for it, since fecal coliforms are the large subset and parts of the microbial colony that need to be tested in drinking water quality. The authors used the term “microbial water quality in this systematic review and meta-analysis study.

11. The sources of drinking water should be specified, noting that water treatment affects bacterial contamination reduction.

Response: Thanks a lot for your eagle eyes! The reviewer is correct. We are very grateful for this implicit comment and suggestion. Thank you so much for such a valuable intellectual input. The authors critically considered this valid input. The newly modified change was highlighted with a red font color in the cleanly revised manuscript.

12. Standards/guidelines for fecal coliform contamination in drinking water should be included.

Response: Thank you so much for pointing this out. The reviewer is flawless. We greatly appreciate your implied observation and recommendation. The authors include standards for fecal coliforms contamination in drinking water. WHO guidelines/Standards and Ethiopian standard agency for fecal coliforms contamination of drinking water were must not be detectable in any 100 ml water sample. The newly modified change was highlighted with red font color in the clean revised manuscript.

13. Change "Drinking water Availability" to "Drinking water availability".

Response: Thanks a lot for your eagle eyes! The reviewer is correct. We are very grateful for this implicit comment and suggestion. Thank you so much for such a valuable intellectual input. The authors critically considered this valid input. The newly modified change was highlighted with a red font color in the cleanly revised manuscript.

14. Ensure consistency in font usage.

Response: Thank you, a lot, for pointing out this! I am really grateful for your insightful comments. The author fully offers the utterly shares the comments and suggestion, which significantly improve the manuscript's quality. We made the font size of the text consistent throughout the whole manuscript. The newly modified change was highlighted with red font color in the clean revised manuscript.

15. Italicize the names of bacterial species: Escherichia, Klebsiella, Enterobacter, and Citrobacter, and Correct "Odd ratio" to "odds ratio".

Response: Thank you, a lot, for pointing out this! Thank you so much for such valuable intellectual input. The Author utterly shares the comments and suggestion which immensely enhance the manuscript quality. The newly modified change was highlighted with red font color in the clean revised manuscript.

16. Change "Study time" to "duration of study".

Response: Definitely! The reviewer is correct. We are grateful for this implicit comment and suggestion. The authors critically reviewed the section. The newly modified change was highlighted with red font color in the clean revised manuscript

17. Specify the full name of the GBI quality parameter.

Response: Absolutely! We really appreciated this important observation. We are at loss for describing your commitments and dedication. The Authors utterly shares the comments and suggestion which immensely enhance the manuscript, subsequently the abbreviations were not present in abstract section. The newly modified change was highlighted with red font color in the clean revised manuscript.

18. Use "fecal coliforms" in plural form.

Response: Thank you very much. The reviewer is perfect. We are grateful for your robust and rigorous comments and suggestions. Hence, the author utterly shares the concerns and comments of the reviewer. Thus, now, with great appreciation, the authors have considered your valid concern and explicitly incorporated your input into the clean revised manuscript, highlighted with a red font color.

Reviewer #2:

This study protocol, entitled systematic review and meta-analysis on the prevalence and contributing factors of fecal coliform contamination in drinking water in Ethiopia addresses a critical public health issue. The methodology that was outlined is comprehensive and follows established guidelines, demonstrating a thorough approach to this important topic. However, I have identified several areas where the manuscript could be strengthened to enhance its clarity, depth, and overall impact. Please find below my detailed comments and suggestions for improvement.

19. The search terms should contain the species of fecal coliform bacteria, including Escherichia, Klebsiella, Enterobacter, and Citrobacter. This can integrate all possible publications conducted in Ethiopia relevant to the exact microbial quality of drinking water in this country.

Response: Thank you very much. The objective of this study mainly focused on the prevalence of fecal coliforms contamination from drinking water in Ethiopia in cumulative way (second stage of te

---

## [Decision Letter · Decision Letter 1]

12 Aug 2024

PONE-D-24-12272R1Prevalence and associated factors of microbial water quality from drinking water in Ethiopia: A Systematic Review and Meta-Analysis ProtocolPLOS ONE

Dear Dr. Worku,

Thank you for submitting your manuscript to PLOS ONE. After careful consideration, we feel that it has merit but does not fully meet PLOS ONE’s publication criteria as it currently stands. Therefore, we invite you to submit a revised version of the manuscript that addresses the points raised during the review process.

We look forward to receiving your revised manuscript.

Kind regards,

D. Daniel, Ph.D.

Academic Editor

PLOS ONE

Journal Requirements:

**Additional Editor Comments:**

Please check a minor comment from the reviewer

Reviewers' comments:

Reviewer's Responses to Questions

**Comments to the Author**

1. Does the manuscript provide a valid rationale for the proposed study, with clearly identified and justified research questions?

Reviewer #1: Yes

Reviewer #2: Yes

2. Is the protocol technically sound and planned in a manner that will lead to a meaningful outcome and allow testing the stated hypotheses?

Reviewer #1: Yes

Reviewer #2: Partly

3. Is the methodology feasible and described in sufficient detail to allow the work to be replicable?

Reviewer #1: Yes

Reviewer #2: Yes

4. Have the authors described where all data underlying the findings will be made available when the study is complete?

Reviewer #1: Yes

Reviewer #2: Yes

5. Is the manuscript presented in an intelligible fashion and written in standard English?

Reviewer #1: Yes

Reviewer #2: Yes

6. Review Comments to the Author

You may also provide optional suggestions and comments to authors that they might find helpful in planning their study.

Reviewer #1: The questions and concerns I addressed are thoroughly explained in the manuscript, making it ready for publication.

Reviewer #2: Your manuscript is well prepared and relevant to objectives. I appreciate to your corrections. Here is a list of some suggestions.

1. I recommend that you review grey literature authored by at least two individuals, given the generally lower quality of these studies.

2. When reaching out to corresponding authors for missing information, I suggest making up to three contact attempts to maximize your chances of receiving a response.

3. To enhance your search results, consider adding additional search terms related to "P" in the PICO statement, such as "occurrence," "detection," "proportion," or "rate," to capture a broader range of research.

7. PLOS authors have the option to publish the peer review history of their article (what does this mean? ). If published, this will include your full peer review and any attached files.

**Do you want your identity to be public for this peer review?** For information about this choice, including consent withdrawal, please see our Privacy Policy .

Reviewer #1: No

Reviewer #2: No

---

## [Author Response · Author response to Decision Letter 2]

29 Aug 2024

I responded in an excellent manner for the reviewer concern, comment and suggestion.

---

## [Decision Letter · Decision Letter 2]

20 Sep 2024

PONE-D-24-12272R2Prevalence and associated factors of microbial water quality from drinking water in Ethiopia: A Systematic Review and Meta-Analysis ProtocolPLOS ONE

Dear Dr. Worku,

Thank you for submitting your manuscript to PLOS ONE. After careful consideration, we feel that it has merit but does not fully meet PLOS ONE’s publication criteria as it currently stands. Therefore, we invite you to submit a revised version of the manuscript that addresses the points raised during the review process.

We look forward to receiving your revised manuscript.

Kind regards,

D. Daniel, Ph.D.

Academic Editor

PLOS ONE

Journal Requirements:

Reviewers' comments:

Reviewer's Responses to Questions

**Comments to the Author**

1. Does the manuscript provide a valid rationale for the proposed study, with clearly identified and justified research questions?

Reviewer #2: Yes

2. Is the protocol technically sound and planned in a manner that will lead to a meaningful outcome and allow testing the stated hypotheses?

Reviewer #2: Yes

3. Is the methodology feasible and described in sufficient detail to allow the work to be replicable?

Reviewer #2: Yes

4. Have the authors described where all data underlying the findings will be made available when the study is complete?

Reviewer #2: Yes

5. Is the manuscript presented in an intelligible fashion and written in standard English?

Reviewer #2: Yes

6. Review Comments to the Author

You may also provide optional suggestions and comments to authors that they might find helpful in planning their study.

Reviewer #2: Your manuscript is well-organized and aligns with the stated objectives. Below are some additional suggestions:

1. Line 67: Please provide the full name of "MDG."

2. Line 70: The term "E. coli" should be written in full at its first mention.

3. Line 84: Clarify whether the bacteria found in North Carolina were fecal coliforms, E. coli, or both.

4. Line 85: "Escherichia coli" should be abbreviated and italicized, as you introduced it earlier in line 70.

5. Lines 89 and 94: Since "TTC" is previously defined in lines 77-78, use the abbreviation throughout.

6. Line 92: The word "and" should not be italicized.

7. Line 103: The term "Scooping" should be written in lowercase.

Study Selection

1. Lines 196 and 222: Consider adding an additional reviewer to resolve any discrepancies in the final decision.

2. Lines 280-281: If the authors choose to use only two reviewers for resolving discrepancies, this should be noted as a potential limitation.

7. PLOS authors have the option to publish the peer review history of their article (what does this mean? ). If published, this will include your full peer review and any attached files.

**Do you want your identity to be public for this peer review?** For information about this choice, including consent withdrawal, please see our Privacy Policy .

Reviewer #2: No

---

## [Author Response · Author response to Decision Letter 3]

22 Sep 2024

The comment and suggestion of reviewer is correct and shared the concern you raised, and we corrected and revised all comments of the manuscript.

---

## [Decision Letter · Decision Letter 3]

29 Oct 2024

Prevalence and associated factors of microbial water quality from drinking water in Ethiopia: A Systematic Review and Meta-Analysis Protocol

PONE-D-24-12272R3

Dear Dr. Worku,

We’re pleased to inform you that your manuscript has been judged scientifically suitable for publication and will be formally accepted for publication once it meets all outstanding technical requirements.

Kind regards,

D. Daniel, Ph.D.

Academic Editor

PLOS ONE

Additional Editor Comments (optional):

Reviewers' comments:

Reviewer's Responses to Questions

**Comments to the Author**

1. Does the manuscript provide a valid rationale for the proposed study, with clearly identified and justified research questions?

Reviewer #2: Yes

2. Is the protocol technically sound and planned in a manner that will lead to a meaningful outcome and allow testing the stated hypotheses?

Reviewer #2: Yes

3. Is the methodology feasible and described in sufficient detail to allow the work to be replicable?

Reviewer #2: Yes

4. Have the authors described where all data underlying the findings will be made available when the study is complete?

Reviewer #2: Yes

5. Is the manuscript presented in an intelligible fashion and written in standard English?

Reviewer #2: Yes

6. Review Comments to the Author

You may also provide optional suggestions and comments to authors that they might find helpful in planning their study.

Reviewer #2: I totally agreed with this manuscript and the protocol is well-generated and feasible. All comments are corrected.

7. PLOS authors have the option to publish the peer review history of their article (what does this mean? ). If published, this will include your full peer review and any attached files.

**Do you want your identity to be public for this peer review?** For information about this choice, including consent withdrawal, please see our Privacy Policy .

Reviewer #2: No

---

## [Editor Report · Acceptance letter]

PONE-D-24-12272R3

PLOS ONE

Dear Dr. Worku,

I'm pleased to inform you that your manuscript has been deemed suitable for publication in PLOS ONE. Congratulations! Your manuscript is now being handed over to our production team.

Kind regards,

on behalf of

Mr D. Daniel

Academic Editor

PLOS ONE